# Platelet-Rich Plasma (PRP) Based on Simple and Efficient Integrated Preparation Precises Quantitatively for Skin Wound Repair

**DOI:** 10.3390/ijms25179340

**Published:** 2024-08-28

**Authors:** Mengjie Qiu, Yating He, Haijie Zhang, Yunquan Zheng, Xianai Shi, Jianmin Yang

**Affiliations:** 1College of Chemistry, Fuzhou University, No. 2 Xueyuan Road, Fuzhou 350108, China; 13675093876@163.com (M.Q.); 221320114@fzu.edu.cn (Y.H.); zhanghaijie0716@163.com (H.Z.); 2Fujian Key Laboratory of Medical Instrument and Pharmaceutical Technology, Fuzhou University, No. 2 Xueyuan Road, Fuzhou 350108, China; shixa@fzu.edu.cn (X.S.); jmyang@fzu.edu.cn (J.Y.); 3International Joint Laboratory of Intelligent Health Care, Fuzhou University, No. 2 Xueyuan Road, Fuzhou 350108, China; 4College of Biological Science and Engineering, Fuzhou University, No. 2 Xueyuan Road, Fuzhou 350108, China

**Keywords:** platelet-rich plasma, optimal preparation method, high recovery rate, platelet-poor plasma, skin restoration

## Abstract

Platelet-rich plasma (PRP) has become an important regenerative therapy. However, the preparation method of PRP has not been standardized, and the optimal platelet concentration for PRP used in skin wound repair is unclear, leading to inconsistent clinical efficacy of PRP. Therefore, the development of standardized preparation methods for PRP and the investigation of the dose-response relationship between PRP with different platelet concentrations and tissue regeneration plays an important role in the development and clinical application of PRP technology. This study has developed an integrated blood collection device from blood drawing to centrifugation. Response surface methodology was employed to optimize the preparation conditions, ultimately achieving a platelet recovery rate as high as 95.74% for PRP (with optimal parameters: centrifugation force 1730× *g*, centrifugation time 10 min, and serum separation gel dosage 1.4 g). Both in vitro and in vivo experimental results indicate that PRP with a (2×) enrichment ratio is the most effective in promoting fibroblast proliferation and skin wound healing, with a cell proliferation rate of over 150% and a wound healing rate of 78% on day 7.

## 1. Introduction

PRP is a concentrated platelet solution obtained by centrifuging whole blood. The beneficial effects of platelets are mediated by the numerous cytokines stored in their cytoplasmic granules, which get released in a normal physiological milieu as well as in response to an external stimulus [1,2]. PRP contains several platelet-related cytokines including platelet-derived growth factor (PDGF), transforming growth factor-beta (TGF-β), insulin-like growth factor (IGF), vascular endothelial growth factor (VEGF), and epidermal growth factor (EGF) [2,3,4]. These substances are essential for cell growth and play a critical role in cell proliferation, migration, and differentiation [5,6,7,8,9]. In recent years, PRP has shown promising results in promoting tissue regeneration and repairing bone defects, treating skin wounds, plastic surgery, and sports injuries [10,11,12,13,14,15]. However, the lack of a standardized method for clinical PRP preparation has resulted in varying compositions and clinical efficacies of PRP, posing challenges for medical professionals in its application [16,17,18,19].

Although the preparation technology for PRP is relatively mature, different preparation methods and techniques each have their inherent limitations. Component blood monomer collection is considered the preferred preparation method [20,21], as it operates in a fully enclosed state, reduces the risk of contamination, and provides a high concentration and purity of platelets, but it requires specialized equipment and is relatively expensive. Blood bag collection preparation [17], as an alternative, is also performed in a closed state, reduces blood waste, and allows for multiple uses from a single collection of PRP, but it has issues such as numerous operational steps, time consumption, and a low platelet recovery rate. Preparation using PRP-specific separation kits is easy to use and allows rapid, convenient, and safe preparation of PRP in an outpatient setting [22,23,24], Centrifugation conditions, such as centrifugal force, time, temperature, and the amount of serum separation gel added, are important factors affecting PRP preparation. When using PRP preparation kits, factors to consider include ease of operation, cost, safety, platelet concentration, and purity; currently, the platelet recovery rate of PRP obtained from commercially available PRP preparation kits is less than 70%, indicating a still relatively low level of platelet recovery. In addition, the sterile conditions of the PRP preparation environment are equally crucial to ensure the safety and effectiveness of the treatment. Although PRP therapy is favored for its minimally invasive nature and potential for biological treatment, the safety of its preparation process and the quality assurance of platelets are indispensable links to ensure the therapeutic effect.

PRP has been used as a regenerative treatment for skin tissue [25,26,27]. When used to repair skin wounds, PRP is usually applied directly to the wound or injected into the skin tissue surrounding the wound [28,29,30]. However, the optimal platelet concentration for PRP has not been determined yet, which hinders the widespread promotion of PRP in clinical treatment [31,32], Some studies suggest that the higher the platelet concentration, the better the therapeutic effect. However, this is only partially true. Other studies indicate that excessively high platelet concentrations may inhibit cell activity and prolong wound healing time [32,33,34,35].

This study developed an optimized method for preparing PRP to achieve a high platelet recovery rate. The characteristics of PPP and PRP with different platelet concentrations were characterized, and their activities in wound repair in vitro and in vivo were evaluated. This research can provide a reference for the study of PRP and PPP in the field of regenerative medicine, as well as guidance for the standardized preparation and application of PRP in clinical practice, aiming to further improve the efficacy of PRP cell therapy and tissue regeneration.

## 2. Results

### 2.1. Effect of RCF and Blood Storage Time on Platelet Recovery Rate

In this study, an integrated process for blood collection and centrifugation was developed (Figure 1A), which is easy to operate and largely avoids product contamination. By adding a serum separation gel, the interference of red blood cells and their lysates is removed and the separation boundary between PPP and PRP is made clearer. It helps to obtain more stable PPP and PRP and reduces the fluctuation of blood-active ingredients caused by human operation. Serum separation gel is critical to the production of PRP as it utilizes the density differences between blood components to isolate PRP. When mixed with whole blood and centrifuged, the components stratify with red blood cells at the bottom, plasma at the top, and platelets and white blood cells in between. This gel improves the recovery rate of the PRP, increasing its clinical efficacy.

To investigate the effects of centrifugal force and blood storage time on platelet recovery rate and platelet enrichment ratio, we conducted a one-way experiment. The results showed that the platelet recovery rate and platelet enrichment ratio increased and then decreased between 1000× *g* and 2000× *g*. At a centrifugation force of 1500× *g*, the platelet recovery rate was 84.76 ± 2.70% and the platelet enrichment ratio was 8.18 ± 0.10 times (Figure 1B). However, the platelet recovery rate and platelet enrichment ratio did not continue to increase with a further increase in centrifugation force, and the platelet recovery rate and platelet enrichment ratio decreased significantly when the centrifugation force was further increased to 2000× *g*. Therefore, we speculate that the optimal centrifugation force may be between 1000× *g* and 2000× *g*.

Blood samples are stored at 4 °C [36]. The platelet recovery rates and platelet enrichment ratios gradually decreased with the prolongation of blood storage time, as shown in Figure 1C. The platelet recoveries in the PRP extracted from blood stored for less than 24 h were more than 80%, with a platelet enrichment ratio of more than 7 times. After 48 h of storage, the platelet recovery rate in PRP was 56.43 ± 1.09%, and the platelet enrichment ratio was approximately 5.93 times. Experimental results show that the shorter the storage time of blood, the better it is for preserving the active components of PRP.

### 2.2. Optimizing the Platelet Recovery Rate of PRP Using Response Surface Methodology

#### 2.2.1. Model Fitting and Variance Analysis

Response surface optimization is a statistical technique used to identify the best combination of variables to optimize a response or meet performance criteria, widely applied in industry, engineering, and research. It involves steps like experimental design, data collection, model fitting, analysis, optimization, and validation. By using mathematical models to describe variable relationships, it reduces the number of experiments, boosts efficiency, and provides a scientific basis for decision-making [37]. We utilized a multifactorial experimental design and selected the centrifugal force, centrifugation time, and the amount of serum separation gel added to optimize the conditions. We conducted 17 experiments following a Box-Behnken design and each experiment was repeated 3 times to estimate the error. Appendix A lists the factors, levels, and results of the experiments, and Appendix A displays the ANOVA results. Our results are also presented in three-dimensional plots of the response surfaces (Figure 2A–C). A regression analysis was performed to test the association between platelet recovery and the three variables. The results were used to generate a quadratic polynomial equation, which is shown below:(1)$$Y=84.4+9.63X_{1}+1.25X_{2}-11.13X_{3}+3.50X_{1}X_{2}-4.75X_{1}X_{3}+5.50X_{2}X_{3}-15.58X_{1}^{2}-0.32X_{2}^{2}-4.07X_{3}^{2}$$

The ANOVA test showed that the regression model was highly significant (*p* = 0.0001 < 0.01). However, the misfit term was not significant (*p* = 0.3381 > 0.05). The values of R^2^ (98.31%), R^2^_Adj_ (96.13%), and C. V. (3.96%) indicate that the experimental results fit the model well (Appendix A). Therefore, the regression equation is considered to be highly credible and well-fitting.

#### 2.2.2. Effect of Variables on Platelet Recovery Rate

The regression coefficients (β) in Appendix A show that centrifugal force and the addition of serum separation gel had a positive and highly significant effect on the platelet recovery rate (*p* < 0.01). Additionally, the interaction of centrifugal force and the addition of serum separation gel, as well as centrifugal force and centrifugation time, had a highly significant effect on the platelet recovery rate (*p* < 0.01).

#### 2.2.3. Model Verification

The optimal culture conditions were determined through regression equations and response surface analysis. The centrifugation force was found to be 1730.39× *g*, the centrifugation time was 10.21 min, and the serum separation gel dosage was 1.41 g. For ease of implementation, we modified the optimal parameters to a centrifugation force of 1730× *g*, a centrifugation time of 10 min, and a serum separation gel dosage of 1.4 g. Validation was performed using the optimal conditions assay. The platelet recovery rate was 95.74 ± 1.03% (Appendix A), which was close to the predicted value. This indicates that our modeling was successful, and the recovery rate of PRP platelets prepared by this method was high.

#### 2.2.4. PRP Stability Analysis

We conducted a preliminary investigation on the stability of PRP, with platelet and leukocyte concentrations as the detection targets. As shown in Figure 2D,E, the concentration of platelets under different storage conditions over time was presented. It can be seen that regardless of whether it is at 4 °C or −80 °C, the concentration of platelets exhibited a decreasing trend. Compared to the concentration at day 0, after 28 d of storage, the platelet concentration decreased by approximately 39% under 4 °C conditions and by about 28% under −80 °C conditions. This indicates that the preservation effect of platelets is relatively better under −80 °C conditions. Under both conditions, the concentration of leukocytes decreased by over 95% compared to the initial concentration after 14 d. The experimental results suggest that there is a negative correlation between platelet concentration and storage time, and freezing conditions are advantageous for maintaining platelet concentration.

### 2.3. Growth Factor Analysis for PRP and PPP

Growth factors play crucial regulatory and promotional roles in tissue repair, regeneration, and development processes [29,30]. We examined the growth factor content of activated and non-activated PRP and PPP.

The levels of PDGF (Figure 3A), EGF (Figure 3B), and VEGF (Figure 3C) were significantly higher in activated PRP (a-PRP) and activated PPP (a-PPP) compared to the whole blood group and the non-activated group. Specifically, a-PRP contained 18.64 times more PDGF, 29.58 times more EGF, and 9.21 times more VEGF than whole blood. Similarly, a-PPP contained 6.84 times more PDGF, 11.21 times more EGF, and 3.12 times more VEGF than whole blood. The levels of PDGF, EGF, and VEGF were significantly higher in activated PRP (a-PRP) and activated PPP (a-PPP) compared to the whole blood group and the non-activated group. These growth factors are released from platelets after activation through the fusion of α-granules with the plasma membrane [38].

Growth factor release in PRP and PPP over 48 h increased with time. At 48 h, PRP released PDGF, EGF, and VEGF at 712.41 pg/mL, 2377.32 pg/mL, and 551.44 pg/mL, respectively. PPP released PDGF, EGF, and VEGF at 405.06 pg/mL, 762.02 pg/mL, and 172.46 pg/mL after 48 h, showing time-dependent increases (Figure 3D–F). In most PRP preparation processes, PPP is discarded to increase the platelet enrichment ratio [39]. Although the PPP by definition contains very low concentrations of platelets, it still represents a reservoir of biologically active molecules and is therefore worth considering for its application.

### 2.4. In Vitro Cellular Effects of PPP and PRP with Different Platelet Concentrations

In this study, we investigated the proliferative effects of PPP, PRP with high platelet concentration, and PRP with low platelet concentration on cells (Figure 4). The platelet concentrations are shown in Appendix A. CCK-8 results (Figure 4B) showed that all drug groups had cell proliferation rates over 100% compared to the control after 24 h, with significant increases for 3T3 cells. PRP (2×) had the best effect, with proliferation rates exceeding 150%, followed by PPP, maintaining rates above 130%. Results suggest both PPP and PRP promote 3T3 cell proliferation, with lower platelet concentration PRP being more effective. Live/dead staining showed minimal dead cells in all groups after 24 h. PRP (2×) had the strongest green signal, indicating high cell viability (Figure 4A). PRP and PPP enhanced cell viability after 48 h. PRP (8×) had a weaker green signal. PRP (2×) was most effective in promoting 3T3 cell migration compared to other groups. After 24 h, migration rates were 100% in PRP (2×) and PPP, 83.22% in PRP (8×), and 34.73% in control (Figure 4C,D). In vitro proangiogenic capacity: all groups formed capillary-like structures after 6 h, with PRP (2×) and PPP showing longer tubes and more nodules. PRP (2×) formed superior vascular networks.

Furthermore, the in vitro proangiogenic capacity of PPP and PRP with different enrichment ratios was evaluated. After 6 h of incubation, all groups showed the formation of capillary-like network structures, in contrast to the control group (Figure 4E). The tube length and the number of nodules were higher in the PRP (2×) group and the PPP group (Figure 4F,G).

### 2.5. In Vivo Evaluation of Wound Healing Efficacy

#### 2.5.1. Wound Appearance Observation and Evaluation

This study evaluated the healing effects of PPP and PRP with different platelet concentrations on experimental wounds in rats. Scabs formed on day 3 in the PRP (2×) and PPP groups, while incomplete scab formation was observed in the control and PRP (8×) groups (Figure 5A). The percentage change in wound area over the 14 d period was shown in (Figure 5B), and the order of wound healing effect was PRP (2×) > PPP > PRP (8×) > control.

#### 2.5.2. H&E and Masson Staining

H&E staining showed PRP (2×) had the highest epithelialization, with the thickest and smoothest epithelium (Figure 5C). PRP (8×) and PPP still had some scab. PRP (2×) had a significantly higher epithelial thickness (108.13 μm) than other groups. PRP (8×) and PPP were above 60 μm, while the control was below 50 μm (Figure 5D). PRP (2×) promotes wound healing and accelerates skin tissue remodeling. The epidermis was dense, dermal tissue was good, and mature epidermal appendages were visible, indicating optimal healing.

Masson’s staining showed PRP (2×) had the highest collagen deposition on day 7, with the most organized fibers (Figure 5E). By day 14, all groups exhibited mature collagen, most orderly in PRP (2×). PRP (2×) had the highest collagen density (57.83%), followed by PPP and PRP (8×) (Figure 5F).

#### 2.5.3. CD31 Staining

A high level of wound angiogenesis promotes wound healing and tissue repair by supplying oxygen and nutrients to the wound site through blood vessels [12]. CD31 staining indicated PRP (2×) promoted the highest blood vessel formation on day 7, surpassing PRP (8×). Drug groups showed significantly higher vascular density than saline (Figure 6A). Quantitative CD31 analysis: PRP (2×) had the highest expression (23.45%), PRP (8×) 10.62%, PPP 14.42%, and saline had the lowest (4.60%) (Figure 6D). PRP (2×) facilitates wound-site angiogenesis, enhancing healing, consistent with in vitro findings.

#### 2.5.4. Collagen I/Collagen III Staining

Collagen I/Collagen III expression (Figure 6B) showed that drug groups exhibited higher fluorescence intensity compared to saline. Particularly, PRP (2×) showed significantly stronger red and green fluorescence than PRP (8×) and PPP. Quantitative analysis of Collagen I, PRP (2×) had 11.95% positive expression, PPP had 5.18%, with no significant difference observed for PRP (8×) and saline. Collagen III quantification (Figure 6F), PRP (2×) had the highest expression (55.68%), followed by PPP (44.59%), PRP (8×) (35.80%), and saline with the lowest. PRP (2×) demonstrates superior collagen generation ability, forming a denser reticular collagen framework.

#### 2.5.5. iNOS Staining

Immunofluorescent staining analysis of iNOS was performed on wound tissues to further evaluate the inflammation status at the wound site (Figure 6C). On the 7th day, compared to the saline group, the proportion of red fluorescence representing iNOS in the drug groups gradually decreased. Quantitative analysis of the proportion of positive iNOS signals is shown in Figure 6G, with the order from largest to smallest being: saline group (42.00%) > PRP (8×) group (33.64%) > PPP group (16.94%) > PRP (2×) group (8.45%). The tissue staining results of the wound site mentioned above indicate that PRP with a lower platelet concentration is more advantageous for wound healing.

## 3. Discussion

PRP has become an important form of regenerative therapy, applied in various fields of regenerative medicine, especially in hair regeneration, wound healing, and sports rehabilitation [5,40,41]. Currently, there is no unified standard for the preparation methods and clinical applications of PRP [16,42]. In this study, a blood centrifugation-integrated PRP preparation device was developed. In one-way experiments, it was determined that short-term storage of blood is more favorable for obtaining platelets. However, the optimal choice of centrifugal force was not determined and interacts with other factors, with centrifugation time and the amount of separating gel added also being factors to be considered. Therefore, we used a Box-Behnken experimental design to select the centrifugal force, centrifugation time, and amount of separating gel added to optimize the PRP experimental conditions. Under the conditions of centrifugation force of 1730× *g*, centrifugation time of 10 min, and addition of 1.4 g of serum separating gel, the recovery of platelets in PRP reached 95.74%. This preparation device reduces the risk of contamination and improves the platelet recovery rate; therefore, in the application of human biological materials, this method may possess higher safety and efficiency.

Prepared PRP and PPP releases are capable of releasing growth factors (PDGF, EGF, and VEGF) that are essential for wound repair [43]. Platelets release more cytokines in PRP than in PPP. In most PRP preparations, PPP is discarded in order to increase the platelet enrichment ratio [17]. a-PPP, however, has significantly higher levels of inflammatory factors and growth factors compared to inactivated PPP. Although the platelet concentration in PPP is very low, it is still a storage reservoir for bioactive molecules [44], so it is worth considering for clinical use.

Some studies have shown that PRP with a low enrichment ratio may lead to unfavorable conditions for wound healing. This could be due to insufficient levels of growth factors in PRP with a low enrichment ratio, which may not effectively activate cell proliferation and promote the repair process [33]. Other studies have indicated that PRP with a high enrichment ratio may not have a significant effect on wound healing. High concentrations of growth factors and proteins may cause excessive cell proliferation or inappropriate cell signaling, thereby interfering with the normal tissue repair process [39].

Through in vitro cellular assays, it was determined that PRP with a low enrichment ratio promotes cell growth more than PRP with a high enrichment ratio. Also, PPP had cell growth-promoting activity. The results of cell dead-viable staining, CCK-8, and cell migration assays indicated that the PRP (2×) group could promote cell proliferation and migration more effectively. The results of the in vitro angiogenesis assay showed that the PRP (2×) group had the best performance in terms of the number of grids and nodes, followed by PPP, while the PRP (8×) group had the least.

In SD rats wound model experiments further validated the superior pro-healing effect of PRP (2×) with a lower platelet concentration in wound healing. Results from H&E and Masson staining showed that PRP (2×) could more effectively promote wound healing and accelerate the structural reconstruction of skin tissue. Blood vessel growth facilitates wound healing and tissue repair [19]. CD31 immunohistochemical staining and quantitative analysis demonstrated that PRP (2×) exhibited superior performance in promoting angiogenesis.

Collagen I and Collagen III can better facilitate wound healing and reduce the likelihood of scarring [45]. Fluorescent double staining of Collagen I/Collagen III showed that both PRP (2×) and PPP had good collagen generation capabilities, which could promote wound healing, the positive expression level of Collagen I reaches 11.95%, while Collagen III reaches 55.68%. Additionally, inducible nitric oxide synthase (iNOS) is responsible for catalyzing the generation of stressful high concentrations of nitric oxide. Pro-inflammatory cytokines first induce the production of large amounts of inducible nitric oxide synthase [46]. iNOS staining results indicated that compared to the PPP group, the wounds in the PRP (2×) group exhibited lower levels of inflammation. PRP with lower concentrations has demonstrated greater activity in promoting cell proliferation and wound healing, which may have significant implications for clinical treatments, especially in the management of skin trauma and burns.

This study holds significant importance for the translatability of human biological materials, indicating the potential for personalized medicine and providing a direction for future research, especially in exploring the effects of different PRP concentrations on skin wound healing. However, to convert these preliminary findings into clinical applications, further clinical trials are necessary to confirm their safety and efficacy, along with the establishment of standardized preparation processes and quality control measures. This will ensure the consistency and reproducibility of PRP therapy, thereby fulfilling its broader clinical treatment potential in the application of human biological materials.

## 4. Materials and Methods

### 4.1. Experimental Animals

Blood was collected by cardiac puncture to obtain a larger volume of blood. The blood was then transferred to a centrifuge tube containing 3.8% sodium citrate (at a ratio of 1:7 by volume to blood) and serum separation gel to extract PRP. Male Sprague–Dawley (SD) rats (weighing 250–300 g) were obtained from Shanghai SLAC Laboratory Animal Co, Ltd. The care and use of SD rats were conducted according to the protocols approved by the National Research Council’s Guide. Healthy male rats were used; rats were initially injected intraperitoneally with sodium pentobarbital (3%, 1 mL/kg). Rats were placed in the supine position. Before sterilizing the anterior chest with povidone-iodine and 75% alcohol, their chests were shaved. Blood was collected from the left side of the chest, specifically from the 3rd and 4th intercostal spaces, where the heartbeat was most intense.

### 4.2. Materials

Serum separation gel was purchased from Wuhan Desheng Biochemical Technology Co., Ltd. (Wuhan, China). A PDGF ELISA kit, EGF ELISA kit, and VEGF ELISA kit were purchased from Shanghai Enzyme-linked Biotechnology Co., Ltd. (Shanghai, China). Calcium chloride (CaCl_2_) was purchased from Sinopharm Chemical Reagent Co., Ltd. (Shanghai, China). DMEM culture medium, penicillin-streptomycin, and phosphate buffer (PBS) were purchased from Shanghai Yuanpei Biotechnology Co., Ltd. (Shanghai, China). CCK-8, live/dead cell staining kits, and Matrigel matrix were purchased from Jiangsu Kaiki Biotechnology Co., Ltd. (Nanjing China). Mouse embryonic fibroblasts (3T3) and human umbilical vein endothelial cells (HUVEC) were purchased from Dingguo Biotechnology Co., Ltd. (Beijing, China). All chemicals were used without further purification.

### 4.3. Factors Affecting the PRP Preparation Process

#### 4.3.1. Effect of Relative Centrifugal Force (RCF) on Platelet Recovery Rate

The centrifugation time was set at 10 min. Whole blood was used, and the amount of serum separation gel was 1 g. PRP was prepared using an L-15 centrifuge (Sigma, Burbank, CA, USA) with different centrifugal force settings—500× *g*, 1000× *g*, 1500× *g*, and 2000× *g*—and then the platelet enrichment ratio and platelet recovery rate were measured.

#### 4.3.2. Effect of Blood Storage Time on Platelet Recovery Rate

With a fixed centrifugal force of 1500× *g*, a centrifugation time of 10 min, and a dosage of 1 g of serum separation gel, PRP was prepared by selecting blood that was stored at the same temperature for different durations (0 h, 24 h, 48 h), and the platelet enrichment ratio and platelet recovery rate were then measured

### 4.4. Optimizing Platelet Recovery Rate of PRP Using Response Surface Methodology

The PRP preparation process required only a single centrifugation step, and we executed all procedures in a benchtop centrifuge. The Box-Behnken design model was established to increase platelet production. After centrifugation, the layer of red blood cells settled under the serum separation gel, a significant number of platelets and white blood cells adhered to the surface of the serum separation gel, while the plasma above was platelet-free. Approximately four-fifths of the PPP was transferred to a new tube for preservation, leaving the platelets and white blood cells attached to the top layer of the serum separation gel. The remaining one-fifth of the PPP was gently vortexed to suspend the platelets and white blood cells, producing PRP. Relative centrifugal force (X_1_), centrifugation time (X_2_), and serum separation gel content (X_3_) were selected as variables for response surface methodology experiments. All variables were set at 3 levels (−1, 0, 1), with X_1_ at 1000, 1500, and 2000× *g*, X_2_ at 5, 10, and 15 min, and X_3_ at 0.5, 1.0, and 1.5 g. The means of the triplicate experiments were fitted with a nonlinear quadratic model:(2)$$Y=\beta_{0}+\beta_{1}X_{1}+\beta_{2}X_{2}+\beta_{3}X_{3}+\beta_{4}X_{1}X_{2}+\beta_{5}X_{1}X_{3}+\beta_{6}X_{2}X_{3}+\beta_{7}X_{1}^{2}+\beta_{8}X_{2}^{2}+\beta_{9}X_{3}^{2}$$
where Y was the response value; X_1_, X_2_, and X_3_ were the independent variables; β_0_ represented the intercept; β_1_ to β_3_, β_4_ to β_6_, and β_7_ to β_9_ were the linear, interaction, and quadratic coefficients, respectively. An analysis of variance (ANOVA) was performed to test the significance of the model using Design Expert 8.0.6 (Stat-Ease, Inc., Minneapolis, MN, USA).

#### PRP Stability Analysis Method

To investigate the stability of PRP, the PRP prepared was evenly divided into 1.5 mL centrifuge tubes. It was divided into 2 groups, each with 3 parallels. The samples were stored according to the experimental design, and the changes in platelet concentration and white blood cell concentration were measured and observed using a BC-5000 animal-specific blood cell analyzer (URIT, Shenzhen, China) within 0 to 28 days.

### 4.5. PRP and PPP Activation and Growth Factor Quantification

PRP was activated with 20 mM CaCl_2_. All samples were incubated at 37 °C for 1 h and then at room temperature for 16 h. To recover the activated PRP (a-PRP), the treated samples were centrifuged at 3000× *g* at 18 °C for 20 min. The supernatant was aspirated to obtain the activated PRP, and all samples were stored at −80 °C. In the preliminary experiments, we found that PPP also contained a small number of platelets, so the same steps were used to activate PPP (a-PPP). Initial plasma, PRP, PPP, a-PRP, and a-PPP samples were analyzed. Concentrations of three growth factors—PDGF, EGF, and VEGF—were measured before and after activation. Growth factor release profiles from PRP and PPP were also measured over a 48-h period at 6 different time points.

### 4.6. In Vitro Activity Studies of PPP and PRP with Different Platelet Concentrations

The prepared PRP was mixed with PPP in different proportions, with V_PRP_:V_PPP_ ratios of 1:0 and 1:4, to obtain high-concentration PRP and low-concentration PRP. In order to determine the effects of PPP and PRP with different platelet concentrations on fibroblast growth, we designed three treatment groups and one blank group (without any treatment) for the experiments:(1)PRP (8×) has an enrichment ratio of 8 to 9 times,(2)PRP (2×) has an enrichment ratio of 2 to 3 times,(3)PPP group,(4)Control group.

#### 4.6.1. Cell Proliferation Assay

3T3 cells in the logarithmic growth phase were taken. 1 mL of trypsin was added and digested for 1 min to obtain a single-cell suspension. The suspension was inoculated into 96-well plates at a density of 2 × 10^3^ cells/well and incubated at 37 °C in a 5% CO_2_ incubator for 24 h. The medium discarded, then serum-free medium mixed with drugs was added, creating 5 parallel wells for each drug group. A blank control group was also included. After a 24 h incubation, the solution was discarded and 10% CCK-8 solution was added to each well. Absorbance at 450 nm was measured using an enzyme marker after a 2 h incubation.

#### 4.6.2. Cell Growth Assay

For live/dead staining, cells were obtained after culturing for one and three days, respectively. Next, 3T3 cells were incubated in PBS containing 1 μM Calcein-AM and 1 μM PI for 30 min. After the incubation, live and dead cells were observed using a TE2000-S inverted fluorescence microscope (Nikon, Tokyo, Japan).

#### 4.6.3. Cell Migration Assay

To investigate the effect of different levels of PRP and PPP enrichment on 3T3 cell apoptosis, 3T3 cells (2 × 10^5^ cells/well) were seeded in 24-well plates and cultured until they formed a confluent monolayer. The monolayer was then scraped in a straight line with a sterile pipette tip, followed by rinsing with PBS to remove any cellular debris. Subsequently, a serum-free medium mixed with drugs was added, with a blank control group being established. Cell migration was monitored by microscopy and quantified using Image J software.

#### 4.6.4. In Vitro Angiogenesis

The matrix gel was placed in 24-well plates and cured at 37 °C for 1 h. A 300 μL volume of HUVEC cell suspension (2 × 10^5^ cells/mL in serum-free medium) was added, followed by the addition of PRP with different enrichment ratios and PPP. After a 6-h incubation, the capillary-like structures of HUVEC cells were examined by microscopy, and the average lumen length and the number of nodules were calculated.

### 4.7. In Vivo Activity Studies of PPP and PRP with Different Platelet Concentrations

To investigate the effects of different enrichment ratios of PRP and PPP on wound healing, 36 male SD rats weighing 200–300 g were utilized, with 9 rats per group. After being injected with a 3% pentobarbital sodium solution at a dosage of 1.0 mL/kg, autologous PRP was used for the treatment of full-thickness skin wounds in the rats. Initially, blood was collected from the rats in the medication groups—PRP (8×) group, PRP (2×) group, and PPP group which was used for the preparation of autologous PRP. The back areas of the rats were then depilated, and two circular wounds were created using a punch biopsy instrument (1 cm in diameter). Saline, PRP (8×), PRP (2×), and PPP were sprayed onto the wound sites, which were then bandaged with sterile gauze after 10 min. The treatments were performed at intervals of 3 days. Optical images of the wounds were captured at 0, 3, 7, and 14 d using a digital camera. The wound size was accurately measured using ImageJ software. And their granulation tissue at the wound, neoplastic, and surrounding tissues was immediately collected. Histological changes were observed through H&E, Masson’s, immunohistochemical staining, and immunofluorescence staining.

### 4.8. Statistical Analysis

All results were repeated three or more times for independent experiments and reported as mean ± standard deviation. Statistical significance between results was determined through one-way ANOVA and Student’s *t*-test. A significance level of *p* ≤ 0.05 was considered statistically significant (ns *p* > 0.05, * 0.01 < *p* ≤ 0.05, ** 0.001 < *p* ≤ 0.01, *** 0.0001 < *p* ≤ 0.001, **** *p* ≤ 0.0001).

## 5. Conclusions

We have successfully developed a user-friendly integrated PRP preparation device for blood collection and centrifugation, significantly reducing the risk of product contamination and achieving a high platelet recovery rate of up to 95%. Experimental results indicate that PRP with lower platelet concentrations exhibits higher activity in promoting cell proliferation and skin wound healing compared to PRP with higher platelet concentrations. Additionally, it was found that components rich in PPP can provide essential nutrients to cells, promoting their growth. This study only focused on exploring two concentrations of PRP, and future work will further investigate the effects of more PRP concentrations on skin wound healing to refine our research.

## Figures and Tables

**Figure 1 ijms-25-09340-f001:**
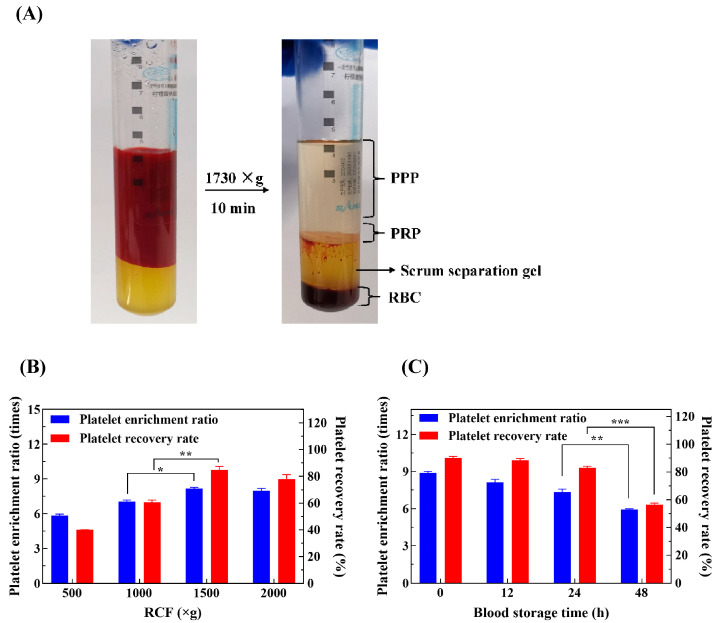
Analysis of the PRP preparation process and factors affecting platelet recovery rate and enrichment ratio. (**A**) PRP preparation process, (**B**) effect of centrifugal force on platelet recovery rate and platelet enrichment ratio, (**C**) effect of blood storage time on platelet recovery rate and platelet enrichment ratio. (* 0.01 < *p* ≤ 0.05, ** 0.001 < *p* ≤ 0.01, *** 0.0001 < *p* ≤ 0.001).

**Figure 2 ijms-25-09340-f002:**
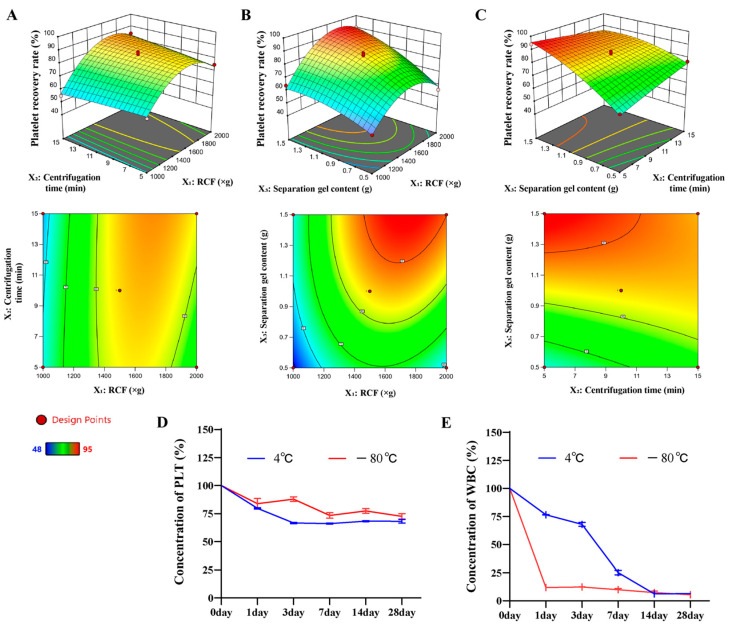
Response surface optimization experiment results and PRP stability analysis. (**A**) Centrifugation force and time, (**B**) centrifugation time and serum separation gel addition, (**C**) centrifugation time and amount of serum separation gel added, (**D**) change of PLT concentration from 0 to 28 d, (**E**) change of WBC concentration from 0 to 28 d.

**Figure 3 ijms-25-09340-f003:**
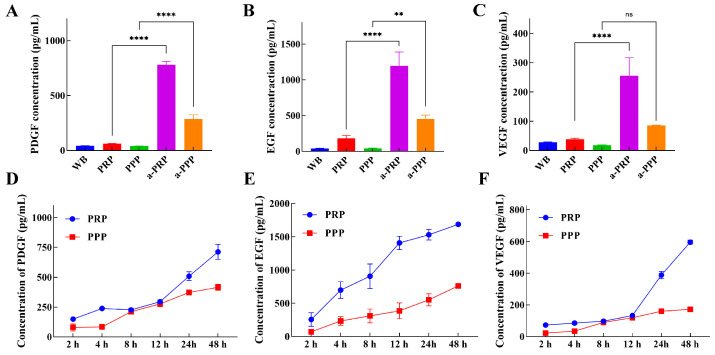
Growth factor analysis for PRP and PPP. (**A**) PDGF concentration, (**B**) EGF concentration, (**C**) VEGF concentration, (**D**) PDGF release curve, (**E**) EGF release curve, (**F**) VEGF release curve. (ns *p* > 0.05, ** 0.001 < *p* ≤ 0.01, **** *p* ≤ 0.0001).

**Figure 4 ijms-25-09340-f004:**
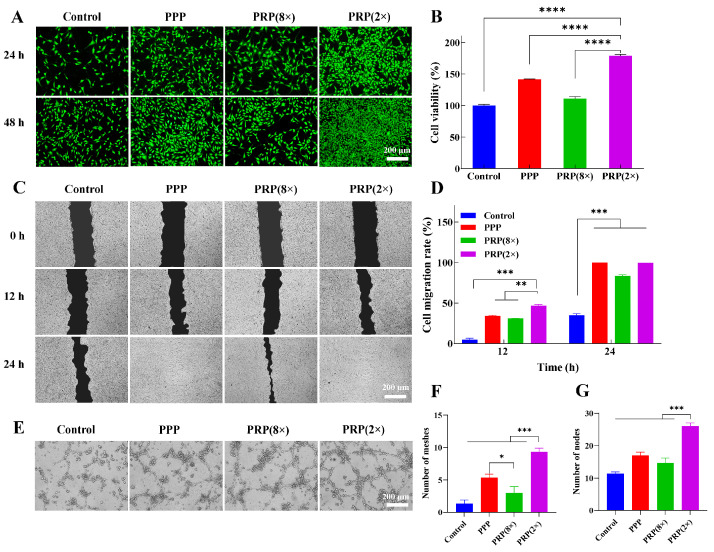
In vitro cellular effects of PPP and PRP with different enrichment ratios. (**A**) Live/dead fluorescence images of 3T3 cells after incubation for 24 h, (**B**) cell proliferation of 3T3 cells after culturing for 24 h before CCK-8 analysis, (**C**) images of 3T3 cells migration, (**D**) cell migration rate, (**E**) in vitro angiogenesis of HUVEC cells, (**F**) the number of meshes, (**G**) the number of meshes. (* *p* < 0.05, ** *p* < 0.01, *** *p* < 0.001, **** *p* < 0.0001).

**Figure 5 ijms-25-09340-f005:**
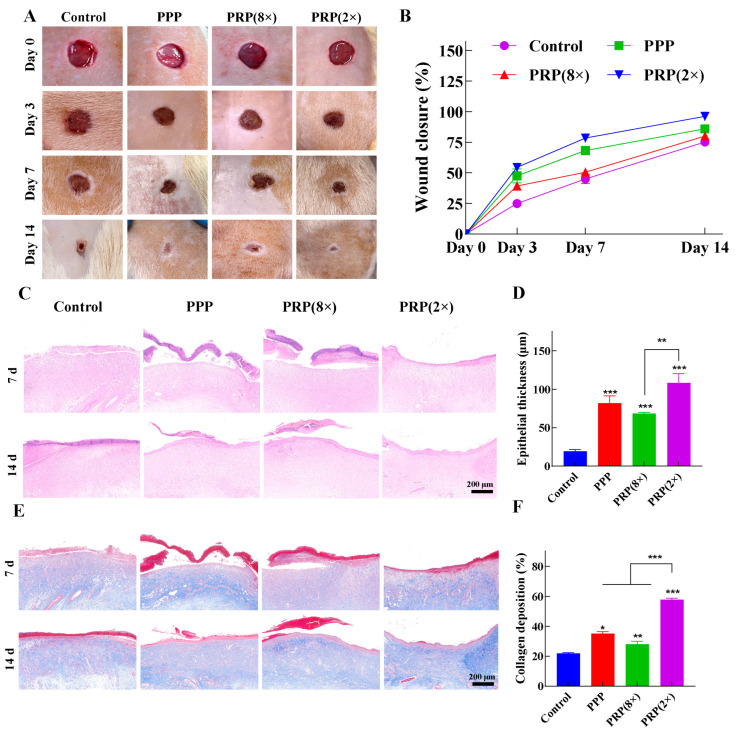
In vivo evaluation of wound healing efficiency (*n* = 9). (**A**) Photographs of wound healing after treatment with different medications, (**B**) wound closure area on days 0, 3, 7, and 14, (**C**) H&E staining images, (**D**) thickness of the dermis in different medication treatment groups, (**E**) Masson staining images, (**F**) collagen deposition in different medication treatment groups. (* 0.01 < *p* ≤ 0.05, ** 0.001 < *p* ≤ 0.01, *** 0.0001 < *p* ≤ 0.001).

**Figure 6 ijms-25-09340-f006:**
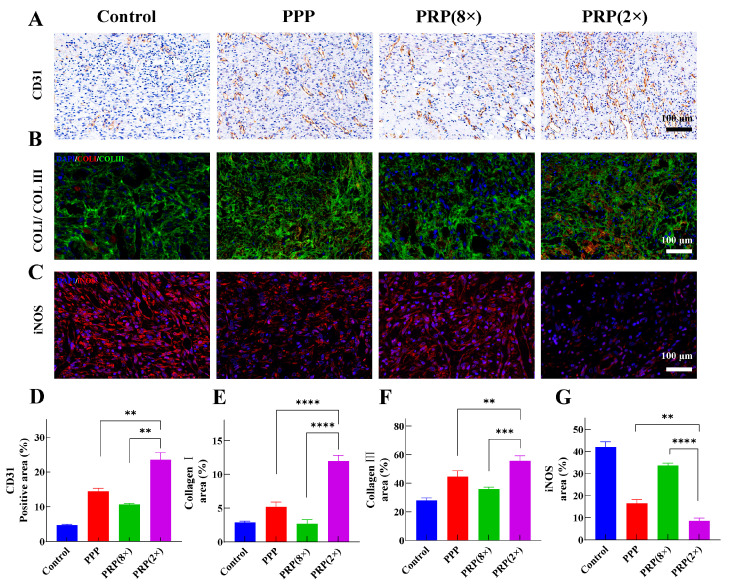
Immunohistochemical and immunofluorescent staining analysis of the skin tissue at the wound site. (**A**) CD31 staining images, (**B**) Collagen I/Collagen III staining images, (**C**) iNOS staining images, (**D**) quantitative analysis of CD31, (**E**) quantitative analysis of Collagen I, (**F**) quantitative analysis of Collagen III, (**G**) quantitative analysis of iNOS. (** 0.001 < *p* ≤ 0.01, *** 0.0001 < *p* ≤ 0.001, **** *p* ≤ 0.0001).

## Data Availability

The original contributions presented in the study are included in the article/Appendix A, further inquiries can be directed to the corresponding author/s.

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
