# Peer review of "Platelet-Rich Plasma (PRP) Based on Simple and Efficient Integrated Preparation Precises Quantitatively for Skin Wound Repair"

_ijms, 2024, doi:10.3390/ijms25179340_

Round 1

Reviewer 1 Report

Comments and Suggestions for Authors

1) Abstract: Authors must be more precise and provide data

2) Introduction: Auhtors should porvide more information related to the state of the art and more refs

3) Results:

-Please, provide more information related to serum separation gel and discuss the state of the art of the separation gels

- Figure 1: the letter a) is missing

- Figure 1 b) and c): units in the "x" axis are missing and

- Figure 1: please provide temperature of blood storage

- Figure 2: what is the concept of "response surface optimization"? Please, explain it

- Line 160: activated PRP: How is the activation and times?

-Figure 4: please, provide the different enrichment ratios

-Figure 5: please provide numbers to PRP high and PRP low

-Figure 5: How many samples/mice were used? and repetitive samples?

4) Discussion: Please, provide more refs and provide more discussion

Comments on the Quality of English Language

English was ok

Author Response

Thank you very much for sending us the review comments on our manuscript (Manuscript Number: ijms-3223578) entitled “Platelet-rich plasma (PRP) based on simple and efficient integrated preparation precises quantitatively for skin wound repair”. We appreciate your kind consideration of this manuscript for possible publication in International Journal of Molecular Sciences after the requested revision.

We appreciate your comments and suggestions, which are very helpful in improving our manuscript. We have carefully revised our manuscript according to your comments and suggestions. The answers and responses to your questions and comments are listed below. The revised parts are highlighted in red in the revised manuscript.

Comment 1: Abstract: Authors must be more precise and provide data

Response 1: Thank you for your insightful comments. We agree with this comment. We have provided more specific data in the abstract to clarify our research findings (in red, lines 21~26).

“Response surface methodology was employed to optimize the preparation conditions, ultimately achieving a platelet recovery rate as high as 95.74% for PRP (with optimal parameters: centrifugation force 1730 ×g, centrifugation time 10 min, serum separation gel dosage 1.4 g). Both in vitro and in vivo experimental results indicate that PRP with a 2(×) enrichment ratio is the most effective in promoting fibroblast proliferation and skin wound healing, with a cell proliferation rate of over 150% and a wound healing rate of 78% on day 7.”

Comment 2: Introduction: Auhtors should porvide more information related to the state of the art and more refs

Response 2: Thank you for your insightful comments. We agree with these comments. We have added more information and additional references (in red) to the introduction of the article.

“Although the preparation technology for PRP is relatively mature, different preparation methods and techniques each have their inherent limitations. Component blood monomer collection is considered the preferred preparation method [20,21], as it operates in a fully enclosed state, reduces the risk of contamination, and provides a high concentration and purity of platelets, but it requires specialized equipment and is relatively expensive. Blood bag collection preparation [17], as an alternative, is also performed in a closed state, reduces blood waste and allows for multiple uses from a single collection of PRP, but it has issues such as numerous operational steps, time consumption, and a low platelet recovery rate. Preparation using PRP-specific separation kits is easy to use and allow rapid, convenient, and safe preparation of PRP in an outpatient setting [22-24]. Centrifugation conditions, such as centrifugal force, time, temperature, and the amount of serum separation gel added, are important factors affecting PRP preparation.”

Comment 3:  Please, provide more information related to serum separation gel and discuss the state of the art of the separation gels

Response 3: We thank the reviewer for the careful review. We have updated the discussion of serum separation gel technology in section “2.1” (in red).

“Serum separation gel is critical to the production of PRP as it utilizes the density differences between blood components to isolate PRP. When mixed with whole blood and centrifuged, the components stratify with red blood cells at the bottom, plasma at the top, and platelets and white blood cells in between. This gel improves the recovery rate of the PRP, increasing its clinical efficacy.”

Comment 4: Figure 1: the letter a) is missing, b) and c): units in the "x" axis are missing, and please provide temperature of blood storage

Response 4: We thank the reviewer for the careful review. We have added the serial numbers of Figure 1 and the units of the X-axis, as shown in Figure 1. In addition, the storage temperature of the blood is 4°C, which has been clarified in section “2.1” (in red). The following literature indicates that the optimal storage temperature for whole blood is 4°C:

Daniel N Darlington, Jacob Chen, Xiaowu Wu, Jeffery Keesee, Bin Liu, Andrew P Cap; Whole Blood Stored at 4°C for 7 Days Is Equivalent to Fresh Whole Blood for Resuscitation of Severe Polytrauma. Blood 2014; 124 (21): 1558. doi: https://doi.org/10.1182/blood.V124.21.1558.1558.

Comment 5: Figure 2: what is the concept of "response surface optimization"? Please, explain it

Response 5: Thank you for your insightful comments. Response surface optimization is a statistical technique widely used in industry, engineering, and scientific research to identify the optimal combination of multiple independent variables that maximizes or minimizes a response variable or meets specific performance criteria. The methodology involves steps such as experimental design, data collection, model fitting, model analysis, response surface analysis, determination of optimal conditions, and validation experiments. By fitting mathematical models to describe the relationships between variables and responses, and using these models for analysis and optimization, response surface optimization helps to reduce the number of experiments, improve efficiency, and provide a scientific basis for decision making. The concept of "response surface optimization" has been added to the section “2.2.1. Model fitting and variance analysis” (in red).

Comment 6: Line 160: activated PRP: How is the activation and times?

Response 6: We agree with this comment. The activation time and method of PRP has been clarified in section "4.5. PRP and PPP activation and growth factor quantification".

“PRP was activated with 20 mM CaCl2. All samples were incubated at 37°C for 1 h and then at room temperature for 16 h. To recover activated PRP (a-PRP), treated samples were centrifuged at 3000 ×g at 18°C for 20 min. The supernatant was aspirated to obtain activated PRP, and all samples were stored at -80°C.”

Comment 7: Figure 4: please, provide the different enrichment ratios

                     Figure 5: please provide numbers to PRP high and PRP low

Response 7: We fully agree with your comments. We have clarified the enrichment ratio of PRP in all figures and throughout the text. Specifically, we have updated the original “PRP(high)” and “PRP(low)” to “PRP(8×)” and “PRP(2×)” respectively (in red), respectively. To accurately reflect the different concentration levels of PRP, the enrichment ratio of PRP is also added in section "4.6. in vitro activity studies of PPP and PRP with different platelet concentrations" (in red). These changes facilitate a more intuitive understanding of the research content for the reader.

Comment 8: Figure 5: How many samples/mice were used? and repetitive samples?

Response 8: Thank you for your insightful comments. The experiment was performed with 36 male SD rats weighing 200-300 g, with 9 rats per group (section "4.7. in vivo activity studies of PPP and PRP at different platelet concentrations" in red).

Comment 9: Discussion: Please, provide more refs and provide more discussion

Response 9: Thank you very much for your insightful comments. Efforts have been made to improve the discussion and more references have been added according to the reviewers' suggestions (section "3. Discussion" in red).

Reviewer 2 Report

Comments and Suggestions for Authors

Manuscript by Qiu et al. addresses the standardization and validation of a method for preparing platelet-rich plasma (PRP). Given PRP's popularity in regenerative medicine and related research, standardizing its preparation is crucial due to the inherent variability of biological isolates and donor variability. This manuscript is highly valuable in this context.

I have several comments that need to be addressed before the manuscript can be accepted for publication:

1.      Image Quality: The quality of the images needs improvement, either through better resolution or larger size, as they are often nearly illegible.

2.      Source of Blood: The manuscript does not clearly specify the origin of the blood used for PRP and other products. If only animal blood was used, it is essential to discuss whether the same procedure is expected to work with human blood. Has this been verified?

3.      Discussion on Human Applicability: The discussion should be expanded to address the translatability of the method to human biological material.

I look forward to reviewing the revised manuscript.

Comments on the Quality of English Language

The text would benefit from stylistic editing but it is fine for readers now. 

Author Response

Thank you very much for sending us the review comments on our manuscript (Manuscript Number: ijms-3223578) entitled “Platelet-rich plasma (PRP) based on simple and efficient integrated preparation precises quantitatively for skin wound repair”. We appreciate your kind consideration of this manuscript for possible publication in International Journal of Molecular Sciences after the requested revision.

We appreciate your comments and suggestions, which are very helpful in improving our manuscript. We have carefully revised our manuscript according to your comments and suggestions. The answers and responses to your questions and comments are listed below. The revised parts are highlighted in red in the revised manuscript.

Comment 1: Image Quality: The quality of the images needs improvement, either through better resolution or larger size, as they are often nearly illegible.

Response 1: We fully agree with your proposals. We have adjusted the clarity and size of the image to make it more recognizable according to journal standards.

Comment 2: Source of Blood: The manuscript does not clearly specify the origin of the blood used for PRP and other products. If only animal blood was used, it is essential to discuss whether the same procedure is expected to work with human blood. Has this been verified?

Response 2: Thank you for your insightful comments. We agree with this comment. In this study, we only used the blood from SD rats for the experiment and did not use human blood. The preparation and use of PRP has been widely used in clinical treatments, but there is currently no standardized protocol. The work we have done provides some theoretical support for the standardized application of PRP, and we will also carry out clinical trials in our subsequent research.

Comment 3: Discussion on Human Applicability: The discussion should be expanded to address the translatability of the method to human biological material.

Response 3: We fully agree with your suggestions. We have discussed the applicability of this method to human biological materials in the "3. Discussion" section (in red).

“In one-way experiments, it has been determined that short-term storage of blood is more favorable for obtaining platelets. However, the optimal choice of centrifugal force has not been determined and interacts with other factors, with centrifugation time and the amount of separating gel added also being factors to be considered. Therefore, we used a Box-Behnken experimental design to select the centrifugal force, centrifugation time, and amount of separating gel added to optimize the PRP experimental conditions. Under the conditions of centrifugation force of 1730 ×g, centrifugation time of 10 min, and addition of 1.4 g of serum separating gel, the recovery of platelets in PRP reached 95.74%. This preparation device reduces the risk of contamination and improves the platelet recovery rate; therefore, in the application of human biological materials, this method may possess higher safety and efficiency.”

“PRP with lower concentrations has demonstrated greater activity in promoting cell proliferation and wound healing, which may have significant implications for clinical treatments, especially in the management of skin trauma and burns.”

“This work holds significant importance for the translatability of human biological materials, indicating the potential for personalized medicine and providing a direction for future research, especially in exploring the effects of different PRP concentrations on skin wound healing. However, to convert these preliminary findings into clinical applications, further clinical trials are necessary to confirm their safety and efficacy, along with the establishment of standardized preparation processes and quality control measures. This will ensure the consistency and reproducibility of PRP therapy, thereby fulfilling its broader clinical treatment potential in the application of human biological materials.”

Round 2

Reviewer 1 Report

Comments and Suggestions for Authors

Authors have adequately addressed the comments

Author Response

Thank you very much for sending us the review comments on our manuscript (Manuscript Number: ijms-3113578) entitled “Platelet-rich plasma (PRP) based on simple and efficient integrated preparation precises quantitatively for skin wound repair”. We thank you for your kind consideration of this manuscript for possible publication in International Journal of Molecular Sciences after the requested revision.

We appreciate your further comments and suggestions, which are very helpful in improving our manuscript. We have carefully revised our manuscript according to your comments and suggestions. The answers and responses to your questions and comments are listed below. The revised parts are highlighted in blue in the revised manuscript.

Comments and Suggestions for Authors: Authors have adequately addressed the comments

Response: We thank the reviewer for appreciation to our manuscript, and the insightful comments.